# Heparinase I treatment to overcome RNA quantification interference in heparinized liver donor samples: One size fits all?

**Mar Dalmau**[ID][1,2]**, Ramón Charco**[2]**, Itxarone Bilbao**[1,2,3]**, Cristina Dopazo**[1,2,3]**, Mireia Caralt**[2]**, José Andrés Molino**[ID][4] **Concepción Gómez-Gavara**[ID][1,2,3,*]

**1** Barcelona Autonoma University, Universitat Autònoma de Barcelona, Barcelona, Spain, **2** Department of HPB surgery and Transplant, Hospital Universitari Vall d'Hebron, Vall d 'Hebron Institute of Research (VHIR), Barcelona, Spain, **3** Centro de Investigación Biomédica en Red de Enfermedades Hepáticas y Digestivas (CIBEREHD), Madrid, Spain, **4** Department of Paediatric Surgery, Hospital Universitari Vall d'Hebron, Barcelona, Spain

\* concepcion.gomez@vallhebron.cat

## Abstract

### Background

MicroRNAs have emerged as potential biomarkers of liver injury during organ transplantation due to their specificity, easy detection and stability in many biofluids. Heparin, which has a well-known inhibitory effect on RT-qPCR based measurements, is commonly used during organ donation. Heparinase I treatment has been used to overcome the inhibiting effect of heparin in RNA RT-qPCR analysis. However, there is a lack of evidence regarding its effective, feasible use improving specific miRNA quantification yield in the liver transplant setting. The aim of this study is to evaluate the effect of heparinase I on miRNA detection levels by RT-qPCR in different samples from liver donors.

### Methods

Prospective, single-centre study including evaluation of liver biopsy, perfusate fluid and serum from deceased organ donors from October 2019 to May 2021. Samples from brain death donors (DBD, n = 4) and donors after circulatory death recovered with abdominal normothermic regional perfusion (DCD n = 4) were analysed for the presence of liver-injury related miRNAs (miR-122 and miR-148a) in the absence or presence of heparinase I (6 IU or 12 IU) to evaluate its effect on miRNA detection levels by RT-qPCR. A subgroup of heparinized serum samples from patients undergoing cardiopulmonary bypass was analysed for validation purposes. The study is registered with ClinicalTrials.gov (NCT06611046), and accrual is complete.

### Results

The expression of miR-122 relative to reference genes was 44.5, 16.8 and 4.2-fold higher in liver biopsy, perfusates and serum respectively, while miR-148a was

Data availability statement: All relevant data are within the paper and its Supporting Information files.

Funding: This project was partially supported by awards from the Spanish Liver Transplant Society (SETH) to RC and MD.

Competing interests: The authors have declared that no competing interests exist.

Abbreviations: aNRP, abdominal normothermic regional perfusion, AST, serum aspartate transaminase, BMI, body mass index, CCI, Charlson Comorbidity Index, CIT, cold ischemic time, CDmiR, cholangiocyte-derived miRNAs, Ct, cycle threshold, DBD, death brain donor, DCD, donation after cardiac death, DWIT, donor warm ischemic time, EAD, early allograft disfunction [32], HDmiR, hepatocyte-derived miRNAs, HEP, heparinized serum, ICU, intensive care unit, IGL-1, Institut Georges Lopez-1, INR, international normalised ratio, miRNAs, microRNAs, NHEP, non-heparinized, POD, post-operative day, RNase, ribonuclease, RT-qPCR, reverse transcription-quantitative polymerase chain reaction, SCS, static cold storage, WLST, withdrawal of life-support-treatment.

3.4, 2.2 and 2.6-fold higher, without differential expression between donor groups (p > 0.05). Heparinase I treatment did not improve PCR results and affected miRNA detection yields in a dose-dependent way with delayed and dispersed Ct values. In highly heparinized DCD serum samples, heparinase I treatment significantly reduced the relative expression of miR-122 and miR-148a compared to non-treated samples, 2-fold and 6.1-fold, p < 0.05 respectively. Moreover, treatment with heparinase I led to a rise in lost values, from 12.5% to 25% in perfusates and 67.7% to 68.7% in serum samples treated with 6IU and 12IU of heparinase I respectively.

## Conclusions

The need for heparinase I treatment to overcome RNA quantification interference in heparinized samples should be addressed in each individual analysis. Heparin inhibition seems variable among miRNAs, and the additional handling with heparinase may affect reliable miRNA quantification due to RNA degradation, introducing bias in gene expression interpretation.

## Introduction

Graft quality assessment at the time of liver transplantation is crucial to ensure early graft recovery, morbidity, and long-term survival [1,2]. To date, organ shortage remains a global issue, and increased use of extended criteria donor grafts as well as machine perfusion for in-situ and ex-situ preservation have both become widespread practice to cope with that [3]. However, this practice demands more objective, dynamic and easily obtainable biomarkers to assess donor liver quality and guide the decision for organ use or discarding before implantation.

MicroRNAs (miRNAs) are short, non-coding RNAs that have emerged as important regulators of gene expression at the post-transcriptional level [4]. They are mainly located intracellularly but can also be found as circulating miRNAs transported by membrane-derived vesicles, proteins, and lipoprotein complexes. Their stability in extreme conditions, such as multiple freeze-thaw cycles, incubation at room temperature for at least 24 hours, and their resistance against serum RNase degradation widens the possibilities of circulating miRNAs as promising non-invasive biomarkers [5,6].

In recent years, significant achievements have been made in revealing the molecular profiles related to poor graft outcomes, which could therefore become novel biomarkers for graft evaluation. In the liver transplant setting, specific miRNAs are reliable markers to evaluate graft injury due to their organ specificity [7,8]. Both miR-148a and miR-122 are liver-derived miRNAs [9], where the latter constitutes 70% of the total miRNA pool in the liver [10]. They are released in serum at the time of liver injury and acute rejection after liver transplantation, even prior to the elevation of AST and ALT in serum [11,12]. Conversely, their expression in liver biopsies is reduced in post-reperfusion biopsies with prolonged warm ischemic times [13]. They

are also released in perfusates that are recovered at the end of cold ischemia time, where miR-122 is high in grafts that develop early allograft disfunction and ischemic biliary complications after transplant [14,15].

Real-time quantitative PCR (RT-qPCR) is the most widely used technique for miRNA analysis due to its high sensitivity and specificity. Although PCR amplification enhances miRNA detection, several factors can impact the accuracy of results, including interference during sample processing, inefficiencies in nucleotide extraction and the inhibitory effects of con-taminants present in the sample. Within these, anticoagulants, such as heparin, have a dose-dependent inhibitory effect in the amplification of miRNA in plasma [16,17]. Prophylactic heparin is routinely administered in organ donors during hospital stay, and especially during liver procurement to prevent blood clotting, which can interfere and lead to inconsis-tencies in miRNA outcome interpretation. The potential inhibitory effect of heparin on miRNA analysis can be counteracted with heparinase I, a bacterial enzyme. Nevertheless, its use has not been standardized and there is scant evidence of its effectiveness in serum, liver tissue and perfusion fluid collected in the liver donor transplant setting [18,19].

As a part of an ongoing research project for identification of miRNA biomarkers predictive for liver-injury in death brain donors (DBD) and circulatory death donors (DCD) before liver transplantation, the present study aimed to evaluate the effectiveness of heparinase I to improve the miRNA quantification yield by RT-qPCR. The primary objective of this study is to evaluate the performance of a commonly used heparinase I dose for the reliable quantification of liver-derived miRNAs (miR-122 and miR-148a) in different liver donor samples. To support the results, a secondary objective aims to evaluate the effect of heparinase I in a subgroup of patients undergoing elective valvuloplasty cardiac surgery, since high heparin dosages are used during cardiopulmonary bypass.

## Methods

### Study population

Eight combined perfusates, liver biopsies and serum samples were collected prospectively from brain death donors (DBD, n = 4) and donors after circulatory death (DCD, n = 4) at Vall d'Hebron University Hospital, Spain, between Octo-ber 2019 and May 2021. Donors for re-transplantation, paediatric and split transplant were excluded to avoid biopsy-related injury [20].

In a separate cohort of four patients without liver disease undergoing elective valve surgery with cardiopulmonary bypass, consecutive serum samples were collected before and after heparinization in April 2021.

Written informed consent was obtained from patients and relatives of donors included in the study. Samples and data from patients were provided by the Vall d'Hebron University Hospital Biobank (PT20/00107), integrated in the Spanish National Biobanks Network, and they were processed following standard operating procedures. The study protocol was approved by the medical ethics committee of the Vall d'Hebron University Hospital [PR(AG)46/2019] and conforms to the principles outlined in the Declaration of Helsinki. The study is registered with ClinicalTrials.gov (NCT06611046), and accrual is complete.

### Sample collection and processing

In the liver donor cohort, blood collection varied according to the type of donor (Fig 1): in DBD donors, a blood sample of 10mL was collected before laparotomy (DBD_T0), which preceded the administration of 300 IU/kg of heparin bolus admin-istered before aortic cannulation. In DCD donors, a first sample of blood was obtained before withdrawal of life-sustaining measures (DCD_T0), which was followed by 300 IU/Kg heparin bolus administration at the onset of donor hypotension (systolic blood pressure < 60 mmHg). Five minutes after asystole, the abdominal normothermic regional perfusion circuit (aNRP) was established by postmortem cannulation of the aorta and abdominal inferior vena cava. A second sample of blood was collected from this heparinized circuit (DCD_T1). The aNPR pump ran for 60–120 minutes for liver viability assessment purposes.

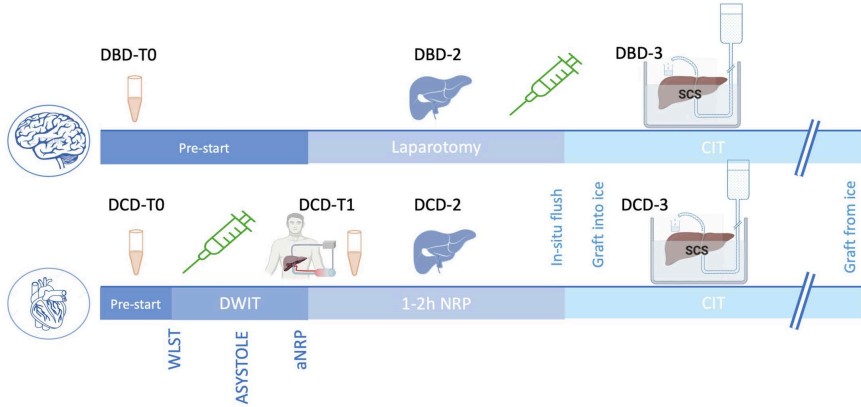

**Fig 1. Sample chronogram collection in DBD and DCD donors.** Timeline of heparin bolus administration (300 IU/kg) marked with a syringe.

Blood samples were collected into serum separation tubes and centrifuged at 2,500 g for 10 minutes, less than four hours after collection. The serum was separated and stored in RNase-free tubes at -80ºC for further analysis. Biopsies of wedge liver tissue or tru-cut were obtained after inspection of the abdominal viscera and stored in RNAlater at -80ºC. Finally, the abdominal organs were flushed with 8 L of cold preservation solution (IGL-1) and then the liver was removed. In the back table, 500 mL of perfusion fluid was flushed into the portal vein and 30 mL of perfusate were recovered from the hepatic veins and stored at -80ºC for further processing.

In the cohort of patients who underwent cardiovascular surgery, a non-heparinized blood sample was obtained after sternotomy, and a second sample of heparinized blood was collected from the cardiopulmonary bypass circuit 30 minutes after administration of a 300 IU/kg bolus of heparin, which was followed by cannulation and cardioplegia.

All samples were analysed for presence of liver-derived miR-122 and miR-148a by RT-qPCR. To normalize the data, two endogenous miRNAs (miR-103a, miR-191) were selected as reference control genes based on their published stability values [13,21,22].

Before miRNA isolation in serum and perfusate samples, haemolysis was analysed by measuring absorbance at 414 nm with a Nanodrop ND2000, and to improve the miRNA yield, 0.8 mg/µL of MS2 carrier was added [ref.#19165948001 from Merck (the Netherlands)]. To monitor quality of miRNA isolation, a 0.02 fmole of UniSp4 spike was added [miRCURY LNA$_{TM}$ RNA Spike-in kit ref.#339390 from Qiagen (Germany)]. RNA was purified from 200 µL of serum and perfusate samples according to the manufacturer's instructions [miRNeasy Serum/Plasma Advanced kit ref.#217204 from Qiagen (Germany)]. In liver tissue, samples between 10–40 mg were lysed using a 20 G syringe and RNA was isolated with miRNeasy Mini kit according to the manufacturer's instructions [ref#217084 from Qiagen (Germany)]. RNA integrity and concentrations were determined using a BioAnalyzer 2100 [Agilent Technologies (The Netherlands)]. All the tissue samples showed a RIN (RNA Integrity Number) value above 5.

Total RNA (5 µL) isolated from serum, perfusates and liver tissue was incubated in the absence or presence of heparinase I (Bacteroides Heparinase I ref#174P0735L from Werfen (Spain) at two different concentrations (6 IU or 12 IU) for 1 hour at 37ºC.

RNA concentration was normalized at 5 ng/µL in the case of liver tissue, according to Bioanalyzer measurements. In the case of serum and perfusion fluid, quantification was not possible due to the expected low concentration of miRNAs and the presence of the MS2 carrier. Subsequently, the same volume of each extracted RNA sample was added to the retrotranscription reaction. RNA was reverse transcribed into cDNA following the manufacturer's instructions (Mircury Rt kit LNATM ref.#339340 from QIAGEN). To monitor the cDNA synthesis reaction and subsequent amplification by qPCR, the

exogen spike-in *Caenorhabditis elegans* miR-39 (Cel-miR-39-3p) [Mircury LNA$_{TM}$ PCR Assay ref.#YP00203952 from Qiagen (Germany)] was added for signs of enzymatic inhibition. To this end, in liver tissue samples, 0.5 µL of spike-in were added to 2 µL of RNA (10 ng) in the RT reaction. For serum and perfusion fluid, 0.5 µL of spike-in were added to 1.6 µL of RNA. After the RT, cDNA served as a template for specific-miRNAs qPCR amplification with locked nucleic acid (LNA) primers and SYBR Green master mix (Mircury LNA$_{TM}$ Sybr Green PCR kit ref.#339346). The specific LNA$_{TM}$ PCR primer sets used are detailed in Table 1. Quantitative PCR was performed in a Light Cycler 480 Real-Time PCR System (Roche, United Kingdom), with amplification curves analysed using the Roche LC software, both for the determination of the cycle threshold (Ct) by the second derivative method and for a melting curve analysis. The Ct threshold was automatically set at 0.2 for all miRNA assays, and the upper Ct limit for reliable detection was set at 40 cycles.

## Statistical analysis

To optimize the quality of data, readings with extreme Ct values (Ct < 11 or Ct > 39) were discarded. The mean of the Cts and the standard deviation for each of the miRNAs in the different samples analysed (technical triplicates) were calculated. To improve the quality of the data analysed, in triplicates with high standard deviation (Ct SD > 0.3), the Ct value furthest from the mean was eliminated, leaving a total of at least two values per miRNA per sample. Likewise, for each miRNA, samples with less than two valid replicates were excluded. To correct for potential RNA input or RT efficiency biases, Ct values were normalised using the average Ct of endogenous references (miR-103a and miR-191). MiRNA relative quantification (RQ) levels were analysed using the Livak et al. method $2^{-\Delta\Delta Ct}$, calculated as follows ΔCt (miRNA Ct target – Ct average of endogenous references) and the difference (ΔΔCt) between comparison groups (ΔCt group A – ΔCt group B [23]. Group comparisons were performed using T-test or U-Mann Whitney (non-parametric) for continuous data and Fisher test for qualitative. P-values < 0.05 were considered significant. Statistical analysis was performed using ``R" (R version 4.2.0 (2022-04-22), Copyright (C) 2022 The R Foundation for Statistical Computing, https://www.R-project.org/).

## Results

### Donor and recipient demographics

The characteristics of donors and transplanted recipients are summarized in Table 2. Overall, the donors were middle-age overweight men (66.7%), with mild macrosteatosis. DBD donors had an average maintenance time of 8 hours from brain death diagnosis to retrieval. DCD donors had short functional and warm ischemic times (< 30 min). The aNRP pump ran a mean of 112.5 ± 47.7 minutes and hepatic transaminase levels remained stable and low (mean AST 100 IU/L and ALT 69.5 IU/L after one-hour). The only factor from the donor variables that was significantly different in DBD, was a higher ICU stay and lower natremia. One liver graft from a DBD was not transplanted due to a suspicious kidney finding during retrieval. Two recipients were listed with HVC cirrhosis with HCC, three with alcohol-related disease, one case of polycystic liver disease, and one case of primary sclerosing cholangitis. The MELD average was low (<15) and there were no significant differences between donors. The CCI average was 11.2 with one case of EAD in a DCD graft. There were no liver-related deaths in an average follow-up of 2.7 years.

**Table 1. Endogen LNA$_{TM}$ PCR primer sets.**

| Primer sets from miRNA PCR Assay Qiagen (Germany) | |
| --- | --- |
| miRNA target genes | hsa-miR-122-5p ref.#YP00205664<br>hsa-miR-148a-3p ref.#YP00205867 |
| miRNA reference genes | hsa-miR-103a-3p ref.#YP00204063<br>hsa-miR-191-5p ref.#YP00204306 |

**Table 2. Donor and recipient clinical information.**

| Donor variables | DBD (n = 4) | DCD (n = 4) | Total (n = 8) | P value |
|---|---|---|---|---|
| Age (years) | 66.5 (±11.47) | 56 (±5.10) | 61.25 (±9.95) | 0.14 |
| BMI (kg/m²) | 29.90 (±7.14) | 28.14 (±3.55) | 29.02 (±5.3) | 0.68 |
| ICU stay (days) | 5.00 (±2.94) | 11.75 (±3.59) | 8.38 (±4.72) | **0.02** |
| Donor maintenance (h) | 7.97 (±0.92) | NA | 7.97 (±0.92) | NA |
| Donor cardiac arrest (min) | 4.75 (±9.50) | 30.5 (±25.04) | 17.63 (±22.29) | 0.10 |
| Sodium (mEq/L) | 155.75 (±7.48) | 142.77 (±1.84) | 149.26 (±8.575) | **0.01** |
| AST (UI/L) | 55.25 (±33.31) | 80.25 (±46.14) | 67.75 (±39.58) | 0.41 |
| Liver macrosteatosis (%) | 5.00 (±7.07) | 12.50 (±13.22) | 8.75 (±10.61) | 0.36 |
| WIT (min) | NA | 21.50 (±4.36) | 21.50 (±4.35) | NA |
| f-WIT (min) | NA | 18.25 (±3.20) | 18.25 (±3.20) | NA |
| Hepatectomy time (min) | 36.75 (±14.33) | 48.75 (±12.20) | 42.75 (±13.89) | 0.25 |
| **Recipient variables** | **DBD (n = 3)** | **DCD (n = 4)** | **Total (n = 7)** | **P value** |
| Age (years) | 59.33 (±5.51) | 53.25 (±13.72) | 55.86 (±10.71) | 0.51 |
| BMI (kg/m²) | 29.35 (±3.47) | 30.44 (±7.07) | 29.97 (±5.41) | 0.82 |
| MELD | 12.67 (±6.03) | 15.25 (±12.18) | 14.14 (±9.39) | 0.75 |
| Total ischemic time (min) | 378.3 (±10.4) | 325.0 (±40.4) | 347.85 (±40.81) | 0.07 |
| Peak AST IU/L (7days) | 396.67 (±216.52) | 1630.50 (±407.07) | 1101.71 (±1200.2) | 0.20 |
| Bilirubin mg/dl (7th POD) | 2.53 (±2.51) | 2.10 (±1.39) | 2.28 (±1.77) | 0.78 |
| INR (7th POD) | 1.01 (±0.09) | 1.17 (±1.05) | 1.17 (±1.04) | 0.54 |
| Length of stay (days) | 23.00 (±9.16) | 11.00 (±3.16) | 16.14 (±8.61) | 0.05 |
| EAD | 0 | 1 (25) | 1 (11.1) | 0.37 |
| CCI | 14.53 (±10.10) | 8.7 (±0) | 11.2 (±11.2) | 0.42 |

Summary of donor and recipient clinical information in donation after cardiac death (DCD) and death brain donors (DBD) liver groups. Donor mainte-nance refers to the time from brain death diagnosis to organ retrieval. Data are presented as n (%) or means (±SD) as appropriate.

## MiRNA expression in liver donors

Two different miRNAs that were described in the literature as being hepatocyte-derived, present in blood, liver tissue and perfusion fluid in liver transplant patients (miR-122 and miR-148a), were selected for further analysis in our study cohort [13–15]. Because stability during storage and isolation is a prerequisite for a good biomarker, we set out to determine whether circulating miRNAs in serum and perfusates remained present and were not degraded. RT-qPCR Ct values were normalised using the average Ct of endogenous references genes reported as stable in liver assays in the litera-ture (miR-103a and miR-191), and miRNA relative quantification (RQ) levels were calculated using the method $2^{-\Delta\Delta Ct}$. In the liver donor cohort, miRNA expression within samples differed highlighting miR-122 expression in liver tissue, being 10.6-fold higher than in serum, and 2.6-fold higher than in perfusion fluid. While miR-148 was expressed in similar propor-tions, being slightly higher in liver tissue (1.3-fold and 1.5-fold higher compared to serum and perfusates respectively). In liver tissue, the expression of miR-122 relative to reference genes was on average 44.5-fold higher, while miR-148a was 3.4-fold higher (Fig 2). In perfusates, miR-122 and miR-148a were 16.8 and 2.2-fold higher than reference genes respec-tively. Mean Ct values for miR-122 showed more than threefold difference over miR-148 in both liver tissue and perfusates (S1 and S2 Tables). In serum samples, although Ct values did not exceed 32, miRNA detection was slightly delayed and dispersed for all the genes tested (S3 Table), and both miR-122 and miR-148, showed a less remarkable relative expres-sion of 4.2 and 2.6-fold compared to reference genes respectively. No significant differences were found between donor groups (p > 0.05) (Fig 2).

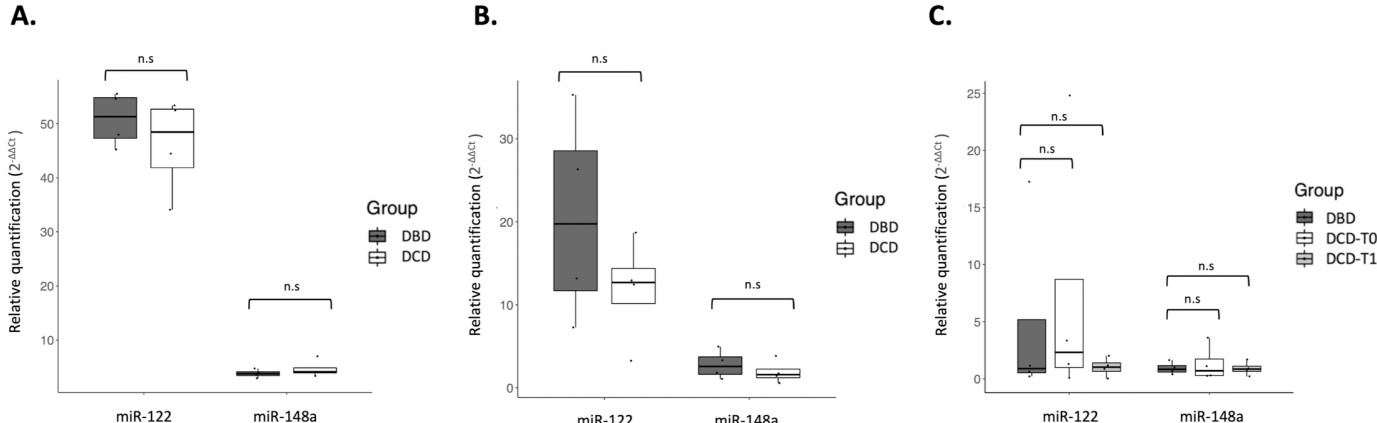

**Fig 2. MiR-122 and miR-148a relative expression in liver donors.** (A) miRNA relative quantification in liver biopsies from DBD (n = 4) and DCD (n = 4). (B) miRNA relative quantification in eight IGL-1 perfusates flushed in the back-table and recovered from the hepatic veins outflow from DBD (n = 4) and DCD (n = 4). (C) miRNA relative quantification in serum samples from DBD donors (n = 4) before laparotomy, and paired serum samples from DCD donors (n = 4) before withdrawal of life-support measures (DCD-T0) and serum from the aNRP circuit (DCD-T1). All boxplots indicate mean ± 95% confidence interval.

It has been widely described that heparin is an enzymatic inhibitor of the retrotranscriptase and the DNA polymerase, so its presence can lead to inaccurate or reduced amplification efficiency, which can negatively affect the results obtained in RT-qPCR experiments. Heparinase I, an enzyme that degrades heparin, has been employed to neutralize this interference, improving miRNA detection, especially in those cases in which targets are less abundant. This increase in sensitivity has been reported in several publications in plasma samples [16,17], as well as in urine and perfusion fluid in the transplant setting [18,19]. Although the Ct values in tissue, perfusion fluid and serum were within analysable ranges, and the Ct values for Cel-miR-39 were consistent and remained stable between the different samples in the study cohort, the potential inhibitory effect of the presence of heparin over RT-qPCR miRNA quantification was still assessed. To this end, eight paired samples of liver tissue, perfusate and serum were incubated in the absence or presence of two different concentrations of heparinase I (6 IU or 12 IU). The effect was assessed using PCR results from endogenous miRNAs (miR-122, miR-148a, miR-103a and miR-191) and the spiked-in UniSp4, a monitor of miRNA isolation. The spiked-in Cel-miR-39 was added as a control prior to the reverse transcription reaction, so was exempt from the effect of heparinase-I.

Conversely, adding this bacterial enzyme in different concentrations in the study cohort did not improve PCR results and affected miRNA detection yields in a dose-dependent way with delayed and dispersed Ct values in almost all the samples (Fig 3 and Table 3): in liver tissue, the relative expression of miR-122 without heparinase I was 1.14-fold (p 0.007) and 1.15-fold (p 0.02) higher compared to those treated with 6IU and 12IU respectively, and miR-148a also showed 1.15-fold (p 0.01) and 1.54 (p 0.007) higher expression in non-treated samples compared to adding 6 IU and 12 IU respectively (S1 Fig). In perfusion fluid, Ct values were progressively delayed and more variable when adding heparinase-I. However, no significant differences were found in relative miR-122 and miR-148a expression with or without treatment. In serum samples collected before heparinization (DBD-T0 and DCD-T0), the relative expression was incomputable due to a significant loss of target and reference miRNA data after treatment with 6 IU and 12 IU heparinase. Nevertheless, the heparinized serum samples in DCD donors collected from the aNRP circuit (DCD-T1) showed a relative expression of miR-148a 6.1-fold (p 0.008) and 5.7-fold (p 0.003) higher without heparinase I compared to those treated with 6 IU and 12 IU respectively, while miR-122 had a 1.4-fold (p 0.44) and 2.0-fold (p 0.04) higher expression without heparinase I compared to those treated with 6 IU and 12 IU respectively (Fig 4).

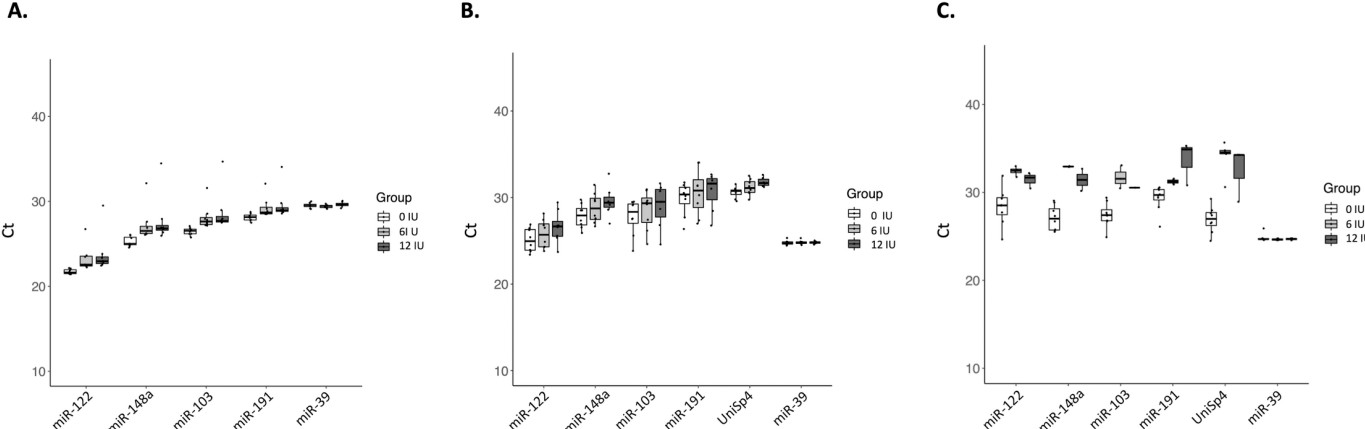

**Fig 3. miRNA Ct values in eight liver donors treated with 0 IU, 6 IU and 12 IU heparinase I.** Ct values from endogenous miRNAs (miR-122, miR-148a, miR-103 and miR-191) and synthetic spiked-in miRNAs (UniSp4 and Cel-miR-39) in (A) liver biopsies, (B) perfusion fluid, and (C) serum from DBD donors (n = 4) before laparotomy, and DCD donors (n = 4) before withdrawal of life-support measures (DCD-T0).

**Table 3. miRNA Ct analysis under different heparinase I concentrations in liver tissue, perfusion fluid and serum from liver donors.**

|  | Heparinase I | Liver tissue |  |  | Perfusion fluid |  |  | Serum |  |  |
|---|---|---|---|---|---|---|---|---|---|---|
|  | dose | n | Ct mean | Ct SD | n | Ct mean | Ct SD | n | Ct mean | Ct SD |
| miR-122 | 0 IU | 8 | 21.72 | 0.29 | 8 | 25.08 | 1.35 | 8 | 28.37 | 2.14 |
|  | 6 IU | 8 | 23.34 | 1.59 | 8 | 25.77 | 1.65 | 4 | 32.42 | 0.51 |
|  | 12 IU | 8 | 23.79 | 2.35 | 8 | 26.61 | 1.82 | 3 | 31.44 | 0.89 |
| miR-148a | 0 IU | 8 | 25.26 | 0.58 | 8 | 27.86 | 1.38 | 8 | 27.08 | 1.43 |
|  | 6 IU | 8 | 27.26 | 2.03 | 8 | 28.81 | 1.73 | 2 | 32.42 | 0.07 |
|  | 12 IU | 7 | 27.74 | 2.78 | 7 | 29.57 | 1.77 | 2 | 31.42 | 1.77 |
| miR-103a | 0 IU | 8 | 27.73 | 2.07 | 8 | 27.73 | 2.07 | 8 | 27.42 | 1.41 |
|  | 6 IU | 8 | 28.54 | 2.26 | 8 | 28.54 | 2.26 | 3 | 31.69 | 1.31 |
|  | 12 IU | 7 | 29.01 | 2.55 | 7 | 29.01 | 2.55 | 1 | 30.53 | NA |
| miR-191 | 0 IU | 8 | 29.86 | 1.89 | 8 | 29.86 | 1.89 | 8 | 29.31 | 1.48 |
|  | 6 IU | 8 | 30.60 | 2.67 | 8 | 30.60 | 2.67 | 2 | 31.22 | 0.46 |
|  | 12 IU | 7 | 30.70 | 2.23 | 7 | 30.70 | 2.23 | 3 | 33.66 | 2.48 |

The spiked-in UniSp4, which was added during miRNA isolation and affected by heparinase-I treatment, shows similar results to the endogenous miRNAs. However, Cel-miR-39 Ct values were stable in all the samples, as it was spiked-in after the heparinase treatment.

Ct means and standard deviations (SD) of endogenous miRNAs (miR-122, miR-148a, miR-103a and miR-191) in liver tissue, perfusion fluid and serum from eight liver donors after coincubation with 0 IU, 6 IU and 12 IU heparinase I before RT-qPCR. (n) number of donor samples analysed in each experiment. Note data lost after heparinase treatment mainly in serum samples. Heparinase treatment affected miRNA detection yields in a dose-dependent way with delayed and dispersed Ct values in all types of samples.

Moreover, treatment with heparinase I had a detrimental effect in the pre-processing data stage increasing extreme Ct values (Ct > 39) in endogenous miRNA triplicates, which are known as NAs (non-available values) (Fig 5). Whereas few NAs were found in liver tissue samples treated with different heparinase concentrations (1%, 2% and 4.2% for 0 IU,

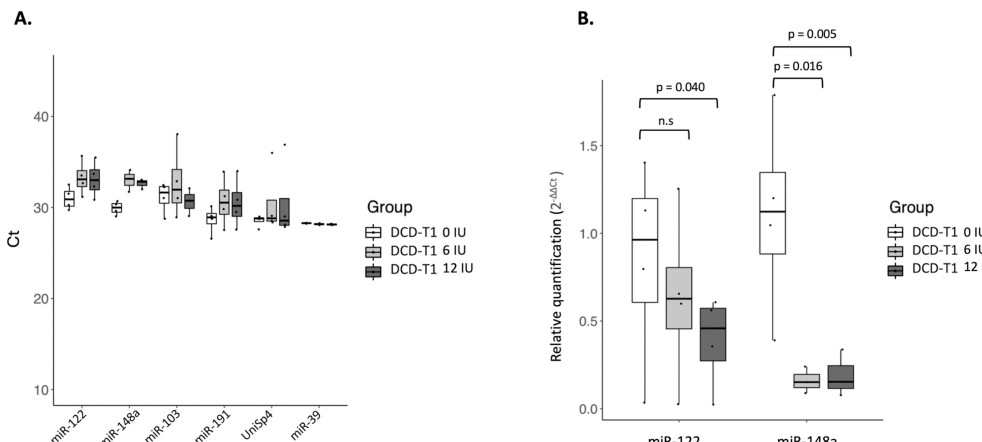

**Fig 4. miRNA analysis in heparinized serum from DCD donors treated with 0 IU, 6 IU and 12 IU of heparinase I.** (A) Ct values of endogenous miRNAs (miR-122, miR-148a, miR-103 and miR-191) and synthetic spiked-in miRNAs (UniSp4 and Cel-miR-39) in four serum samples from the aNRP circuit (DCD-T1) without (0 IU) and treated with 6 IU and 12 IU heparinase I. (B) Heparinase I treatment significantly reduced the relative expression of miR-122 and miR-148a compared to non-treated samples.

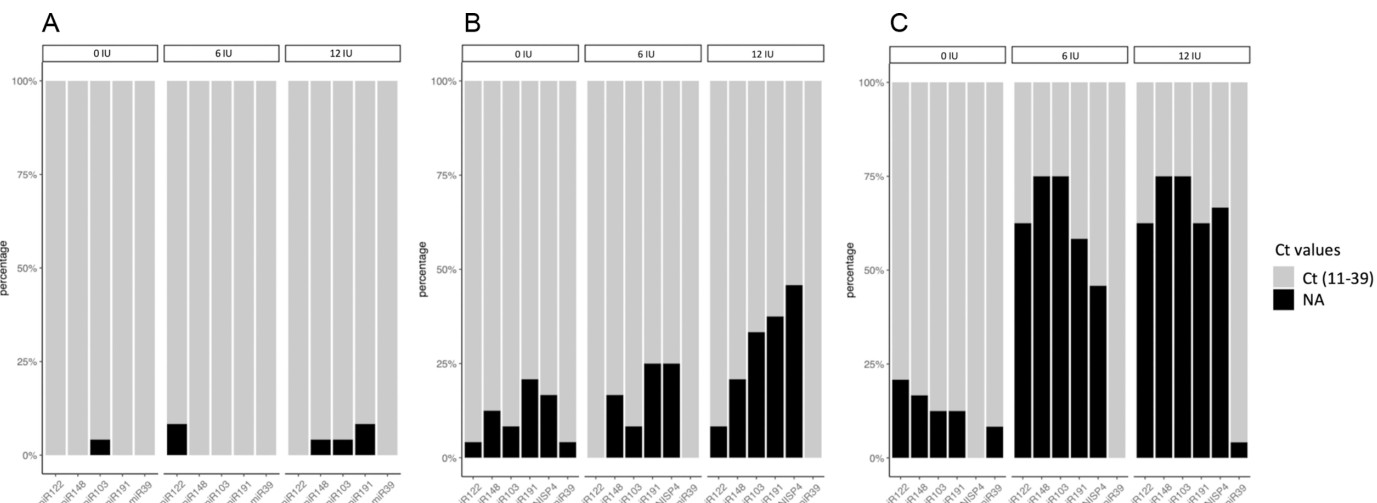

**Fig 5. Percentage of lost data in the pre-processing stage with different heparinase I concentrations.** The percentage of non-available values (NAs), extreme Ct values (<11 or >39) present in each of the miRNA triplicates analysed is shown graphically. NAs in liver tissue samples (A), perfusion fluid (B) and DCD-T0 and DBD-T0 serum samples (C) after treatment with 0 IU, 6 IU and 12 IU of heparinase I. miRNA targets: miR-122 and miR-148a. miRNA reference genes: miR-103a and miR-191. Synthetic spiked-in for miRNA isolation assessment: UniSp4. Synthetic spiked-in miRNA for quality RT-qPCR assessment: Cel-miR-39.

6 IU and 12 IU heparinase I respectively), the impact on perfusates and serum was remarkable. In perfusion fluid, treatment with 12 IU doubled the NAs compared to 0 IU and 6 IU heparinase I (25% Vs 11.4% and 12.5% respectively), and in serum, treatment with 6 IU and 12 IU of heparinase I resulted in 67.7% and 68.7% of lost values respectively. Even so, the spike-in UniSp4 added into perfusates and serum before heparinase treatment underwent a significant loss of 45.8% and 66.7% values after 12 IU heparinase-I coincubation respectively. In contrast, the spike-in Cel-miR-39 detection levels remained unaltered because was added posteriorly, thus had no influence from the heparinase treatment.

## Heparinase I treatment in serum from cardiac surgery patients

To corroborate the effect of heparinase I treatment in a controlled setting, a cohort of four patients without liver-disease who underwent elective cardiac surgery with cardiopulmonary bypass was analysed. All patients were middle-aged males with normal liver function tests and history of valvulopathy (S4 Table). RNA samples of non-heparinized serum collected after sternotomy, and heparinized serum from the bypass system 30 minutes after administration of 300 IU/kg of heparin, were co-incubated with or without 6 and 12 IU of heparinase I during cDNA synthesis. Subsequently, RT-qPCR was performed for the endogenous miRNAs (miR-122, miR148a, miR-103a and miR-191). Detection levels in both non-heparinized and heparinized samples did not improve after heparinase I treatment (Fig 6), and the range of Ct values was strongly increased with approximately one to five cycles (Table 4). In addition, there was a notable loss of values in those samples treated with heparinase I, with 18.7% NAs in non-heparinized samples and 34.4% NAs in heparinized ones.

Ct means and standard deviations (SD) of endogenous miRNAs (miR-122, miR-148a, miR-103a and miR-191) from four non-heparinized and heparinized paired serum samples of patients undergoing cardiac surgery after coincubation with 0 IU, 6 IU and 12 IU heparinase I. (n) number of donor samples analysed in each experiment. Note data lost after heparinase treatment.

## Discussion

The present study showed that miR-122 was highly expressed in liver donor cells and was released into the perfusion fluid flushed through the liver even in the back-table, once the graft is retrieved. Hence, quantification of liver-specific miR-122 after procurement in perfusates could provide a liver quality statement before implantation and assist guidance on liver graft selection. Conversely, miR-122 and miR-148a, which are known to be released in serum at the time of liver injury,

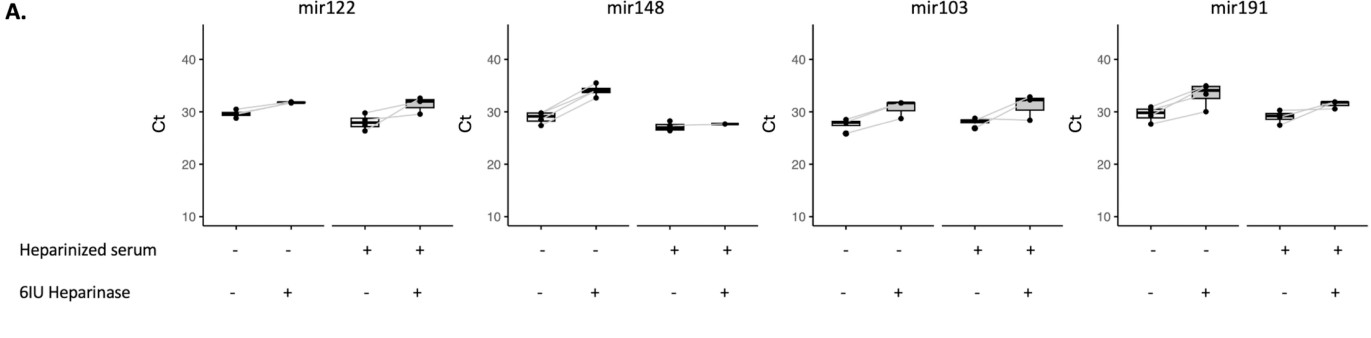

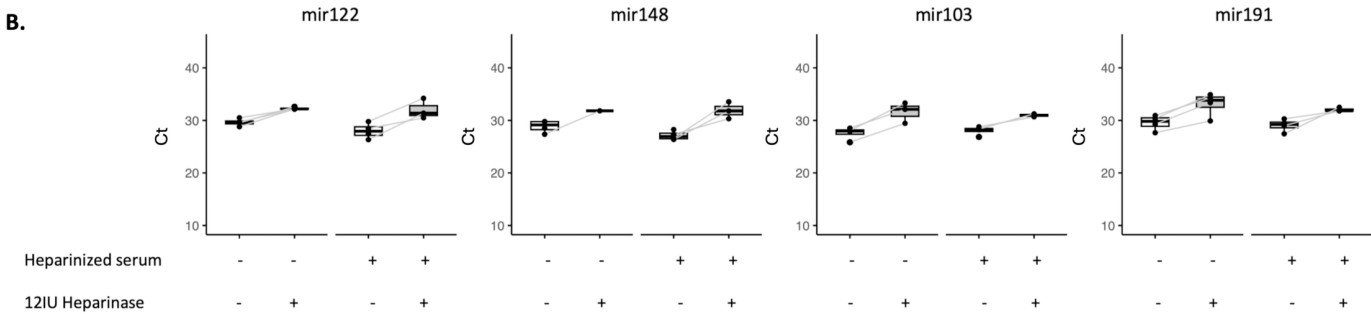

**Fig 6. miRNA Ct analysis in patients undergoing cardiac surgery.** Ct values of endogenous miRNAs (miR-122, miR-148a, miR-103a and miR-191) in non-heparinized and heparinized serum from patients undergoing elective cardiac surgery with cardiopulmonary bypass without heparinase I compared to 6 IU (A) and 12 IU (B) of heparinase I treatment.

**Table 4. miRNA Ct analysis under different heparinase I concentrations in cardiac surgery patients.**

| | Heparinase I dose | Non-heparinized serum | | | Heparinized serum | | |
|---|---|---|---|---|---|---|---|
| | | n | Ct mean | Ct SD | n | Ct mean | Ct SD |
| miR-122 | 0 IU | 4 | 29.62 | 0.70 | 4 | 28.00 | 1.47 |
| | 6 IU | 3 | 31.78 | 0.11 | 3 | 31.39 | 1.61 |
| | 12 IU | 4 | 32.27 | 0.21 | 3 | 32.02 | 1.94 |
| miR-148a | 0 IU | 4 | 28.86 | 1.15 | 4 | 27.13 | 0.86 |
| | 6 IU | 4 | 34.07 | 1.16 | 1 | 27.67 | NA |
| | 12 IU | 1 | 31.82 | NA | 3 | 31.89 | 1.61 |
| miR-103a | 0 IU | 4 | 27.58 | 1.19 | 4 | 28.06 | 0.83 |
| | 6 IU | 3 | 30.70 | 1.72 | 3 | 31.17 | 2.43 |
| | 12 IU | 3 | 31.62 | 1.98 | 2 | 30.97 | 0.39 |
| miR-191 | 0 IU | 4 | 29.55 | 1.44 | 4 | 29.05 | 1.20 |
| | 6 IU | 4 | 33.29 | 2.28 | 3 | 31.43 | 0.75 |
| | 12 IU | 4 | 33.12 | 2.24 | 3 | 32.04 | 0.39 |

were detected in a similar range as refence genes in the study cohort. This highlights the need to investigate a wider expression of liver-derived miRNAs in serum in a larger, heterogenous liver donor cohort including extended criteria grafts, to evaluate their specificity in detecting prompt liver damage. Although heparin is a well-known inhibitor of RT-qPCR, and liver donor biofluids are prone to have traces, treatment with heparinase I has not improved miR-122 and miR-148a detection in serum, perfusates or liver tissue in the study cohort. These results were contrasted in highly heparinized serum samples in a controlled cohort of patients undergoing valve surgery, where treatment with heparinase I did not enhance detection either. In addition, treatment with heparinase I seemed to have a detrimental impact on the pre-processing data, with a significant loss of values in serum and perfusates compared to non-treated samples, and with a global delay in cycle threshold detection for all the miRNAs tested.

The use of heparinase I to overcome the inhibitory effect of heparin has not been standardized [24–26] and there is scant evidence regarding its effectiveness in serum, perfusion fluid and liver tissue collected in the liver donor transplant setting, which requires further discussion.

PCR-based measurements of circulating miRNAs in blood need to be interpreted with caution, as this biofluid is more susceptible to contamination with higher doses of heparin during hospitalization or interventional procedures. While the benefit of heparinase I treatment has been recognized to improve RNA quantification by qPCR in plasma samples [16,17,26,27], its effectivity seems to be heterogenous among different miRNAs. This could be explained by a selective inhibitory effect of heparin on endogenous circulating miRNAs. Boeckle et al. analysed plasma from 11 patients before a cardiac catheterization procedure at 10 and 60 minutes after heparin was administered (bolus 5000 IU heparin; maintenance dose 2500–5000 IU). Only half of the cardiovascular-related miRNAs (miR-133a, miR-34a, miR-378 and miR-499) were profoundly reduced by 10 minutes after heparin administration ($p < 0.05$), while miR-1, miR-92a, miR-126, miR-17 and miR-145 expression were not significantly altered. Moreover, only a third remained significantly reduced at 60 minutes (miR-34a, miR-133a, and miR-208), suggesting that heparin presence influences the detection of only specific miRNAs, and this was sample-time related from heparin addition [16]. This temporary inhibitory effect is likely explained by the short half-life of unfractionated heparin in circulation with disappearance of heparin inhibition between 4 h and 6 h post-administration in plasma samples of cardiac catheterization patients [28,29].

Cohelo-Lima et al. evaluated the effect of in vitro addition of serial doses of heparin and heparinase I to RNA samples with cardiac-enriched miRNAs expression (miR-133b and miR-1). Expression of miR-133b was significantly reduced by almost all tested doses of heparin (0.005 to 2IU) and effectively restored with heparinase I (0.05 to 1 IU). In contrast,

higher supraphysiological in vitro doses of heparin were required to significantly inhibit miR-1 expression, and therefore heparinase treatment had no significant effect in restoring miR-1 Ct values in heparin-contaminated patient samples collected at 30 minutes post-primary percutaneous coronary intervention [30]. The mechanism behind this selective effect of heparin on different miRNAs with variable grades of inhibition is unknown, and future studies are needed to investigate how heparin interacts with different miRNAs and its circulating lipoprotein-transport complexes.

Regarding miRNA analysis in perfusates in the transplant setting, cholangiocyte-derived miRNAs (CDmiR-30e, CDmiR-222 and CRmiR-296) and hepatocyte-derived miRNAs (HDmiR-122 and HDmiR-148a) have been analysed in graft preservation fluid collected at the end of cold ischemia time. These showed that miRNAs in perfusates were stable and protected against RNase degradation by interacting with proteins, and that the HDmiRs/CDmiRs ratios were significantly higher in perfusates obtained from grafts which developed biliary complications after liver transplantation [15]. The same group re-examined the miRNA profiles in a smaller cohort of perfusates to investigate the presence of heparin contamination using 6 IU of heparinase I, which significantly increased the detectability of HDmiR-148a and CDmiR-296. However, no improvement was observed for HDmiR-122 and CDmir-222, suggesting that miRNA detection with heparinase I was more marked in some samples depending on the amount of heparin contamination [19]. Consistently, a high expression of miR-122 and miR-148a was also found in perfusion fluid in the present study, which supports their potential use as non-invasive liver-derived biomarkers. Nevertheless, no differences were noted after heparinase I treatment with either 6 IU or 12 IU. This could equally be attributed to low levels or absence of heparin contamination, probably diluted after 8 L of abdominal perfusion, or a probably low miR-122 and miR-148a sensitivity to heparin inhibition.

Turning now to miRNAs analysis in transplant-related liver tissue, miR-22 levels in post-reperfusion biopsies have been associated with primary non-function after DCD liver transplant [31]. Furthermore, expression of miR-122 and miR-148a in liver tissue has been found to be downregulated with prolonged warm ischemia times [13]. Our results showed that miR-122 expression in liver cells was 10.6-fold higher than serum and 2.6-fold higher than perfusates, without improving detection following heparinase I treatment, suggesting that miRNA analysis in liver biopsies is not influenced by heparin. These results are supported by those reported from Roest et al. in kidney biopsies from living donors, where they demonstrated that heparin had a low or absent interference on RT-qPCR in renal tissue as there were no differences after heparinase I treatment [18].

As expected, in this study miRNA expression in serum and perfusates was lower compared to liver biopsies, due to miRNA liver cell injury release diluted into these biofluids. Therefore, extreme Ct values (Ct > 39) were missed in serum and perfusates miRNA analysis. Despite that, our data comparing control and heparinase-treated samples revealed that the additional manipulation performed during the heparinase treatment caused not only a 0.8 to 5.3-fold Ct delayed detection, but also a loss of 4.2%, 25% and 68% undetectable miRNA values in pre-processing triplicates in liver biopsies, perfusion fluid and serum respectively. This event has been previously reported by Kondratov et al, who demonstrated successful quantification of cardiovascular-related miRNAs (miR-1, mir-208) by using heparinase I treatment in heparinized plasma from patients undergoing coronary artery-bypass-graft surgery. Interestingly, they also proved that the additional handling during heparinase I treatment caused a decrease of 1.5 to 6.5-fold measures of different miRNA targets [28]. There is no clear explanation for this negative effect on miRNA detection after coincubation with heparinase. A possible reason is RNA degradation by remaining RNases in serum and perfusion fluid activated during heparinase coincubation at 37ºC. Less likely, the heparinase preparation could contain RNase activity, given that the enzymes used were purified by a series of column chromatography steps to remove any possible DNA or RNA contamination, being >95% pure as determined by intact mass spectrometry analysis. Even so, the use of heparinase should be considered in each single miRNA experiment, as this extra handling process may affect reliable miRNA quantification introducing bias in gene expression interpretation.

Cel-miR-39 is a synthetic miRNA, absent in mammalian species, commonly used to monitor RT-qPCR efficiency. This exogen miRNA is known to be very sensitive to heparin inhibition, being almost undetectable under small amounts of unfractionated heparin (0.01 IU) [16,18]. In the present study, Cel-miR-39 was not coincubated with heparinase, as was

added posteriorly, and the Ct values in all the samples tested were detectable in optimal threshold ranges. This indicates that the potential traces of heparin remaining in the samples were not sufficient to inhibit the RT and PCR reactions, and/or that the enzymatic reactions could have had a certain grade of tolerance to residual heparin traces. Consequently, as there were no significant signs of heparin inhibition, and moreover, all the miRNAs tested were detected in convincing Ct thresholds, heparinase treatment is considered technically unnecessary in this assay.

This study presents some limitations. Firstly, it was assumed that samples collected after the administration of heparin bolus were highly heparinized, such as blood specimens from the aNRP circuit in DCD donors, or at least contained heparin traces, such as liver biopsies in DCDs and perfusion fluid in both donor groups. As heparin levels were not measured, different types of samples were collected at various stages during donor procurement to ensure a wide range of specimens representing the whole process of liver graft donation before and after administration of heparin, reflecting the reality of the donation setting. Moreover, the results were reproducible with paired samples of non-heparinized and heparinized serum from cardiac surgery patients. Secondly, the study cohort is small due to sample collection constraints, and the endogenous miRNAs tested represent a modest selection of liver-specific injury miRNAs and reference genes for normalization. A high-throughput screening panel of miRNAs across a spectrum of liver disorders tested in a larger donor cohort would provide extended knowledge for further miRNA biomarker research in liver transplantation. Finally, plasma samples were not collected, which could have given varying results compared to serum in terms of heparin miRNA inhibition and effect of heparinase I treatment.

Overall, while there are numerous encouraging reports on the use of specific circulating miRNAs as biomarkers in the transplantation setting, it is important to recognize the factors that might affect the measurement of miRNA levels and take these explicitly into account for objective clinical implementation.

In conclusion, this study addressed methodological hurdles to accurate quantification of liver-derived miRNAs in liver biopsies, serum, and perfusion fluid in liver donors. The aforementioned results have implications for circulating miRNA studies in the liver transplant setting, and particularly in ex-situ machine perfusion techniques, where anticoagulation therapy is administered. In addition, they reinforce the evidence of liver-derived miR-122 and miR-148a expression circulating in biofluids such as perfusion fluid and serum, which could provide information about liver graft quality before implantation. Finally, it can be argued that heparinase I treatment in heparin-contaminated samples should be addressed in each individual miRNA analysis, since heparin inhibition seems variable among miRNAs, which is potentially affected by dose and sampling times, and attempts to remove it may affect reliable miRNA quantification due to RNA degradation, introducing bias in gene expression interpretation.

## Supporting information

**S1 Table. miRNA Ct values in donor liver biopsies.**
(DOCX)

**S2 Table. miRNA Ct values in donor perfusion fluid.**
(DOCX)

**S3 Table. miRNA Ct values in donor serum.**
(DOCX)

**S4 Table. Clinical characteristics and raw data Ct values of patients undergoing cardiac surgery with cardiopulmonary bypass.**
(DOCX)

**S1 Fig. Donor miRNA relative expression with or without heparinase I treatment.**
(DOCX)

## Acknowledgments

We thank the High Technology Unit-UAT, Biobank and Statistics and Bioinformatics Unit from Vall Hebron Institute of Research (VHIR) for the methodological support (Rosa Prieto, RP and Isabel Novoa, IN), technical execution (Pilar Mancera, PM and María Molinos, MM) and bioinformatics and statistics analysis of the qPCR data (Mireia Ferrer, MF). We also thank Brian Brennan for valuable English correction, and Esther Delgado for her involvement and dedication to liver transplant patients.

## Author contributions

**Conceptualization:** Mar Dalmau, Ramón Charco, Itxarone Bilbao, Concepción Gómez-Gavara.

**Data curation:** Mar Dalmau, Cristina Dopazo, Mireia Caralt, Jose Andrés Molino, Concepción Gómez-Gavara.

**Formal analysis:** Mar Dalmau, Concepción Gómez-Gavara.

**Funding acquisition:** Mar Dalmau, Ramón Charco.

**Investigation:** Mar Dalmau, Ramón Charco, Concepción Gómez-Gavara.

**Methodology:** Mar Dalmau, Ramón Charco, Concepción Gómez-Gavara.

**Project administration:** Mar Dalmau, Ramón Charco.

**Resources:** Mar Dalmau.

**Supervision:** Mar Dalmau, Ramón Charco, Itxarone Bilbao, Concepción Gómez-Gavara.

**Validation:** Mar Dalmau, Ramón Charco, Concepción Gómez-Gavara.

**Visualization:** Mar Dalmau, Cristina Dopazo, Mireia Caralt, Jose Andrés Molino, Concepción Gómez-Gavara.

**Writing – original draft:** Mar Dalmau.

**Writing – review & editing:** Mar Dalmau, Ramón Charco, Itxarone Bilbao, Cristina Dopazo, Mireia Caralt, Jose Andrés Molino, Concepción Gómez-Gavara.

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
