## [Decision Letter · Decision Letter 0]

20 Jan 2025

PONE-D-24-48749Heparinase I treatment to overcome RNA quantification interference in heparinized liver donor samples: one size fits all?PLOS ONE

Dear Dr. Gómez-Gavara,

Thank you for submitting your manuscript to PLOS ONE. After careful consideration, we feel that it has merit but does not fully meet PLOS ONE’s publication criteria as it currently stands. Therefore, we invite you to submit a revised version of the manuscript that addresses the points raised during the review process.

We look forward to receiving your revised manuscript.

Kind regards,

Tianwen Wang, Ph.D.

Academic Editor

PLOS ONE

Journal Requirements:

“This project was partially supported by awards from the Spanish Liver Transplant Society (SETH) to RC and MD.”

4. We note that your Data Availability Statement is currently as follows: “All relevant data are within the manuscript and in Supporting Information files.”

5. We note that there is identifying data in the Supporting Information file <S4 table.docx>. Due to the inclusion of these potentially identifying data, we have removed this file from your file inventory. Prior to sharing human research participant data, authors should consult with an ethics committee to ensure data are shared in accordance with participant consent and all applicable local laws.

-Location data

Reviewers' comments:

Reviewer's Responses to Questions

**Comments to the Author**

1. Is the manuscript technically sound, and do the data support the conclusions?

Reviewer #1: Yes

Reviewer #2: Yes

2. Has the statistical analysis been performed appropriately and rigorously? 

Reviewer #1: Yes

Reviewer #2: Yes

3. Have the authors made all data underlying the findings in their manuscript fully available?

Reviewer #1: Yes

Reviewer #2: Yes

4. Is the manuscript presented in an intelligible fashion and written in standard English?

Reviewer #1: Yes

Reviewer #2: Yes

5. Review Comments to the Author

Reviewer #1: This study is interesting. But there are some problems, which must be solved before it is considered for publication.

1. There are many formatting errors or irregularities in the manuscript, such as the lack of spaces between some numbers and units; irregularities such as RNAse should be RNase (line 395); Many references are incomplete, e.g. missing page numbers (ref 7, 21,22…).

2. The image resolution is subpar, with unclear annotations detracting from the overall clarity and coherence of the visual representation. (Fig.2, 3,4,5,6)

3. Tables should be formatted consistently and in a more scientific three-line table format

Reviewer #2: The study aimed to evaluate the effectiveness of heparinase I in improving miRNA quantification yield by RT - qPCR in liver - donor samples, as heparin is an inhibitor in RT - qPCR and heparinase I is used to counteract it, but its use lacks evidence in this context. There are several points to note in this study. 1. heparin levels were not measured, but samples at different stages were collected to represent the whole process. 2. the study cohort is small, and the tested miRNAs and reference genes are a modest selection. A larger cohort with a high - throughput screening panel of miRNAs would be better. 3. plasma samples were not collected, which might lead to different results compared to serum in terms of heparin miRNA inhibition and heparinase I treatment effect.

6. PLOS authors have the option to publish the peer review history of their article (what does this mean? ). If published, this will include your full peer review and any attached files.

**Do you want your identity to be public for this peer review?** For information about this choice, including consent withdrawal, please see our Privacy Policy .

Reviewer #1: No

Reviewer #2: No

---

## [Author Response · Author response to Decision Letter 0]

6 Mar 2025

February 24 2025

PlosOne

Re: Manuscript ID PONE-D-24-48749

Dear Academic Editor and reviewers,

Find enclosed the response to Journal requirements and reviewers’ evaluation.

Journal Requirements:

1.Please ensure that your manuscript meets PLOS ONE's style requirements, including those for file naming.

Thanks for the comment. The manuscript meets PLOS ONE’s style requirements.

Thanks for the comment. Funding-related text has been removed from the manuscript.

“This project was partially supported by awards from the Spanish Liver Transplant Society (SETH) to RC and MD.”

We have included this amended Role of Funder statement in the cover letter. Thanks for changing the online submission form on our behalf.

4. We note that your Data Availability Statement is currently as follows: “All relevant data are within the manuscript and in Supporting Information files.”

We have added all raw data required to replicate the results of the study in Supporting Information files (S1, S2, S3 and S4 tables). Note that the sample name column data has been previously anonymized.

5. We note that there is identifying data in the Supporting Information file <S4 table.docx>. Due to the inclusion of these potentially identifying data, we have removed this file from your file inventory. Prior to sharing human research participant data, authors should consult with an ethics committee to ensure data are shared in accordance with participant consent and all applicable local laws.

The first patient column with anonymous identification data has been removed from the <S4 table.docx> file as follows. Please reconsider adding <S4 table.docx> file again in our file inventory, as we have removed sensible patient information, and we have add a new table with miRNA raw data Ct values.

Patient Age (y) Sex BMI AST (IU/L) ALT(IU/L) Bilirubin mg/dl

1 75 Male 25.31 27 27 0.64

2 54 Male 30.04 18 17 -

3 72 Male 31.56 19 10 0.62

4 52 Male 24.06 25 29 0.62

To this table version:

Age (y) Sex BMI AST (IU/L) ALT(IU/L) Bilirubin (mg/dl)

75 Male 25.31 27 27 0.64

54 Male 30.04 18 17 -

72 Male 31.56 19 10 0.62

52 Male 24.06 25 29 0.62

Reviewers' comments to the Author:

Reviewer #1:

This study is interesting. But there are some problems, which must be solved before it is considered for publication.

1. There are many formatting errors or irregularities in the manuscript, such as the lack of spaces between some numbers and units; irregularities such as RNAse should be RNase (line 395); Many references are incomplete, e.g. missing page numbers (ref 7, 21,22…).

Thanks for the comment. All the formatting errors, irregularities and incomplete references have been amended and highlighted in the “Revised Manuscript with Track Changes” document.

2. The image resolution is subpar, with unclear annotations detracting from the overall clarity and coherence of the visual representation. (Fig.2, 3,4,5,6).

Thanks for the comment. The image resolution has been optimized and annotations such as p-values > 0.05 have been summarized as “n.s” for non-significant to clarify the overview.

All figure files have been upload to the Preflight Analysis and Conversion Engine (PACE) digital diagnostic tool, https://pacev2.apexcovantage.com/ to ensure that figures meet PLOS requirements.

3. Tables should be formatted consistently and in a more scientific three-line table format

Thanks for the comment. All the tables have been formatted as recommended.

Reviewer #2:

The study aimed to evaluate the effectiveness of heparinase I in improving miRNA quantification yield by RT - qPCR in liver - donor samples, as heparin is an inhibitor in RT - qPCR and heparinase I is used to counteract it, but its use lacks evidence in this context. There are several points to note in this study.

1. Heparin levels were not measured, but samples at different stages were collected to represent the whole process.

This study has been conducted as a part of an ongoing research project for identification of miRNA biomarkers predictive of liver-injury in brain death donors and donors after circulatory death before liver transplantation. To ensure a correct methodology, we aimed to evaluate the effectiveness of heparinase I to improve the miRNA quantification yield by RT-qPCR in a small cohort of patients. The primary objective of this pre-study is to evaluate the performance of a commonly used heparinase I dose for the reliable quantification of liver-derived miRNAs (miR-122 and miR-148a) in different liver donor samples.

Prophylactic heparin is routinely administered in organ donors during hospital stay, and especially during liver procurement before vascular cannulation to prevent blood clotting (300 IU/kg). It is well described that heparin can interfere in RT-qPCR and we assumed that our samples (biopsy, serum and perfusion fluid) could be contaminated by different grades of heparin. Given that heparinase-I has already been demonstrated to overcome the inhibitor effect of heparin in in-vitro and controlled in vivo heparinized samples of plasma and perfusion fluid (1-3), and that donor samples could contain variable traces of heparin, our goal was to confirm the valuable effect of heparinase I at common dosages (6 IU and 12 IU) in different donor samples collected during the whole procurement process to improve miRNA quantification yield by RT-qPCR.

Moreover, to support the results obtained, the secondary objective aimed to evaluate the effect of heparinase I in a subgroup of patients undergoing elective valvuloplasty cardiac surgery, since high heparin dosages are used during cardiopulmonary bypass. RNA samples of non-heparinized serum collected after sternotomy, and heparinized serum from the bypass system 30 minutes after administration of 300 IU/kg of heparin, were co-incubated with 6 and 12 IU of heparinase I during cDNA synthesis. This cohort guarantees that the heparinase I treatment is proved over paired samples of serum without heparin and with heparin contamination, regardless of the specific dose.

(1) Coelho-Lima J, Mohammed A, Cormack S, Jones S, Das R, Egred M, et al. Overcoming Heparin-Associated RT-qPCR Inhibition and Normalization Issues for microRNA Quantification in Patients with Acute Myocardial Infarction. Thromb Haemost. 2018 Jul;118(7):1257–69.

(2) Roest HP, Verhoeven CJ, de Haan JE, de Jonge J, IJzermans JNM, van der Laan LJW. Improving Accuracy of Urinary miRNA Quantification in Heparinized Patients Using Heparinase I Digestion. J Mol Diag. 2016 Nov;18(6):825–33.

(3) Verhoeven CJ, Farid WR, de Ruiter PE, Hansen BE, Roest HP, de Jonge J, et al. MicroRNA profiles in graft preservation solution are predictive of ischemic-type biliary lesions after liver transplantation. J Hepatol. 2013 Dec;59(6):1231–8.

2. the study cohort is small, and the tested miRNAs and reference genes are a modest selection. A larger cohort with a high - throughput screening panel of miRNAs would be better.

The study cohort consists of n=8 patients with three to four different samples per patient, and tests two miRNA targets as well as two miRNA reference genes. The study population and miRNA sets are a modest selection because this study is simply a proof of concept to validate a correct methodology of an ongoing research project for identification of miRNA biomarkers for liver injury in liver donors. To support the results obtained, heparinase I effect was tested in another cohort of four patients undergoing cardiac surgery collecting serum samples before and after heparinization. Given that heparinase I treatment did not enhance miRNA detection and seemed to have a detrimental impact on the pre-processing data in both cohorts (donors and patients undergoing cardiac surgery), we consider that these are robust results to recommend that heparinase I treatment in heparin-contaminated samples should be addressed in each individual miRNA analysis.

3. plasma samples were not collected, which might lead to different results compared to serum in terms of heparin miRNA inhibition and heparinase I treatment effect.

In the present study, the uncertain inhibiting effect of heparin on miRNA qPCR detection and non-benefit effect of heparinase-I treatment shown in serum samples might be partially explained by the different nature of serum compared to plasma. While most macromolecular elements of plasma and serum are the same, and both lack blood cells, unlike serum, plasma retains the coagulation cascade clotting factors and fibrinogen. Therefore, exogenous anticoagulants, such as heparin, are retained in plasma bound to the enzyme inhibitor antithrombin III, hence their absence or minor presence in serum samples (4-6). However, although most of the literature describing the effectiveness of heparinase-I to overcome heparin qPCR inhibition refers to plasma samples, there are also studies that support this effect in serum samples (7). For this reason, we considered it relevant to test the heparinase effect in serum, as it is a common medium of sample collection and miRNA analysis. Moreover, similar results have been observed in the perfusion fluid, which supports the results observed in serum in the present study.

(4) Chuang YJ, Swanson R, Raja SM, Olson ST. Heparin enhances the specificity of antithrombin for thrombin and factor Xa independent of the reactive center loop sequence. Evidence for an exosite determinant of factor Xa specificity in heparin-activated antithrombin. J Biol Chem. 2001;276(18):14961-14971.

(5) Björk I, Lindahl U. Mechanism of the anticoagulant action of heparin. Mol Cell Biochem. 1982; 48(3): 161–182.

(6) Plieskatt JL, Feng Y, Rinaldi G, Mulvenna JP, Bethony JM, Brindley PJ. Circumventing qPCR inhibition to amplify miRNAs in plasma. Biomark Res. 2014 Jul;2(1):13.

(7) Chen CC, Peng CC, Fan PC, Chu PH, Chang YS, Chang CH. Practical Procedures for Improving Detection of Circulating miRNAs in Cardiovascular Diseases. J Cardiovasc Transl Res. 2020;13(6):977-987.

---

## [Decision Letter · Decision Letter 1]

30 Mar 2025

Heparinase I treatment to overcome RNA quantification interference in heparinized liver donor samples: one size fits all?

PONE-D-24-48749R1

Dear Dr. Gómez-Gavara,

We’re pleased to inform you that your manuscript has been judged scientifically suitable for publication and will be formally accepted for publication once it meets all outstanding technical requirements.

Kind regards,

Tianwen Wang, Ph.D.

Academic Editor

PLOS ONE

Additional Editor Comments (optional):

Reviewers' comments:

Reviewer's Responses to Questions

**Comments to the Author**

1. If the authors have adequately addressed your comments raised in a previous round of review and you feel that this manuscript is now acceptable for publication, you may indicate that here to bypass the “Comments to the Author” section, enter your conflict of interest statement in the “Confidential to Editor” section, and submit your "Accept" recommendation.

Reviewer #1: All comments have been addressed

Reviewer #2: All comments have been addressed

2. Is the manuscript technically sound, and do the data support the conclusions?

Reviewer #1: Yes

Reviewer #2: Yes

3. Has the statistical analysis been performed appropriately and rigorously? 

Reviewer #1: Yes

Reviewer #2: Yes

4. Have the authors made all data underlying the findings in their manuscript fully available?

Reviewer #1: Yes

Reviewer #2: Yes

5. Is the manuscript presented in an intelligible fashion and written in standard English?

Reviewer #1: Yes

Reviewer #2: Yes

6. Review Comments to the Author

Reviewer #1: (No Response)

Reviewer #2: The author addressed the issues I was concerned about and enhanced the manuscript's quality through revisions. Overall, the manuscript appears to meet publication requirements.

7. PLOS authors have the option to publish the peer review history of their article (what does this mean? ). If published, this will include your full peer review and any attached files.

**Do you want your identity to be public for this peer review?** For information about this choice, including consent withdrawal, please see our Privacy Policy .

Reviewer #1: No

Reviewer #2: No

---

## [Editor Report · Acceptance letter]

PONE-D-24-48749R1

PLOS ONE

Dear Dr. Gómez-Gavara,

I'm pleased to inform you that your manuscript has been deemed suitable for publication in PLOS ONE. Congratulations! Your manuscript is now being handed over to our production team.

Kind regards,

on behalf of

Dr. Tianwen Wang

Academic Editor

PLOS ONE